# Temperature-Dependent Development of *Nitidula rufipes* (Linnaeus, 1767) (Coleoptera: Nitidulidae) and Its Significance in Estimating Minimum Postmortem Interval

**DOI:** 10.3390/insects14030299

**Published:** 2023-03-20

**Authors:** Gengwang Hu, Liangliang Li, Yi Guo, Chengtao Kang, Yinghui Wang, Yanan Zhang, Zhixiang Zhang, Jiangfeng Wang, Yu Wang

**Affiliations:** Department of Forensic Medicine, Soochow University, Ganjiang East Road, Suzhou 215000, China

**Keywords:** forensic entomology, postmortem interval, Coleoptera, *Nitidula rufipes*, in vivo measurement, development, instar determination

## Abstract

**Simple Summary:**

The decay of human or animal carcasses usually attracts the colonization of large numbers of car-rion-related arthropods, especially Diptera and Coleoptera. Since insects are ectotherms, their colonization and development on carcasses are strongly temperature-dependent and predictable, rendering them a powerful tool for estimating the postmortem interval. Nitidulidae is a large family of common storage and agricultural pests of forensic significance. In this study, the development of the sap beetle *Nitidula rufipes* (Linnaeus, 1767) was observed under seven constant temperatures between 16 and 34 °C to describe the growth and morphology as well as survival rates of the different developmental stages using multiple scientific imaging and statistical techniques. We showed that *N. rufipes* can complete its development at temperatures between 16 and 34 °C, where the longest developmental time is 71.0 ± 4.4 days at 16 °C and the shortest is 20.8 ± 2.4 days at 34 °C. Body length measurements combined with morphological characteristics allowed for larval aging and instar discrimination, thus providing relatively complete developmental data for *N. rufipes*. This information serves to aid forensic investigators in determining the postmortem interval of carcasses using *N. rufipes*.

**Abstract:**

Coleoptera, including the family Nitidulidae, are valuable for estimating long-term postmortem intervals in the late stage of body decomposition. This study showed that, under seven constant temperatures of 16, 19, 22, 25, 28, 31, and 34 °C, the developmental durations of *Nitidula rufipes* (Linnaeus, 1767) from oviposition to eclosion were 71.0 ± 4.4, 52.9 ± 4.1, 40.1 ± 3.4, 30.1 ± 2.1, 24.2 ± 2.0, 21.0 ±2.3, and 20.8 ± 2.4 days, respectively. The morphological indexes of body length, the widths of the head capsules, and the distance between the urogomphi of the larvae were measured in vivo. The regression model between larval body length and developmental durations was simulated for larval aging, and the head capsule width and the distance between the urogomphi at different instars were cluster-analyzed for instar discrimination. Based on the developmental durations, larval body length and thermal summation data were obtained, and the isomorphen diagram, isomegalen diagram, linear thermal summation models, and curvilinear Optim SSI models were established. The lower developmental threshold and thermal summation constant of *N. rufipes* evaluated by the linear thermal summation models were 9.65 ± 0.62 °C and 471.40 ± 25.46 degree days, respectively. The lower developmental thresholds, intrinsic optimum temperature, and upper lethal developmental threshold obtained by Optim SSI models were 10.12, 24.15, and 36.00 °C, respectively. The study of the immature stages of *N. rufipes* can provide preliminary basic developmental data for the estimation of minimum postmortem interval (PMI_min_). However, more extensive studies are needed on the effects of constant and fluctuating temperatures on the development of *N. rufipes*.

## 1. Introduction

The natural transformation of human or animal carcasses from highly concentrated organic material to inorganic material to achieve material cycling requires the process of decay, which attracts large numbers of carrion-related arthropods, notably Diptera and Coleoptera [1]. Since insects are all ectotherms, their colonization and development on carcasses are strongly temperature-dependent and predictable, making them a powerful tool in forensic entomology to estimate PMI_min_. Dipterans are often used as a tool to estimate PMI_min_ due to their earlier colonization and frequent occurrence on carcasses [2,3,4]. Forensic entomology, however, is a discipline that predominates in estimating long-term PMI_min_, and in the late stage of body decomposition when Diptera has often left the body by pupation or eclosion, usually only Coleoptera can be found as insect evidence [5,6]. It is often reported that Nitidulidae colonizes in the active and advanced decay stages of the carcass [7,8,9], which means that this family is valuable in forensic entomology study and application [10,11].

The family Nitidulidae (sap beetles) consists of more than 4500 species [12]. They have a complex diet and can feed on flowers, plant sap, fungi, fermented and rotten animal tissues, and plant tissues [13,14,15,16]. They serve as pollinators but also spread pathogenic microorganisms that damage crops and trees [17,18]. Due to their small size and frequent occurrence in stored products [19,20], and with the expansion of international trade, this family is distributed worldwide and considered invasive [21,22,23]. At present, studies on Nitidulidae mainly focus on their role as storage pests, in customs quarantine, and agricultural disease control [24,25,26,27], where they are regarded as pests with economic significance.

There are only a few necrophagous or saprophagous Nitidulidae species that can be found on carcasses. In 38 succession studies and case reports that identified insect species, there was a total of 13 Nitidulidae species, all of which belong to the 4 genera of *Omosita* Erichson, 1843; *Nitidula* Fabricius, 1775; *Carpophilus* Stephens, 1830; and *Glischrochilus* Reitter, 1873 [14,28,29]. There are also a large number of succession studies or case reports listing the family Nitidulidae, but with no specific species identified [2,30,31,32,33,34,35,36,37,38,39,40,41,42,43,44,45,46,47,48,49,50,51,52,53,54,55,56,57,58,59,60,61]. This reflects the family’s frequent association with carcasses. Despite this, the value of Nitidulidae in forensic investigation is largely underestimated [62,63]. At present, the study of Nitidulidae in forensic medicine is limited to taxonomy (mainly the identification of adults and mature larvae). Torres et al. [64] and Cao et al. [65] described the morphology of mature larvae and adults of *Omosita colon* (Linnaeus, 1758); Perris et al. [66] and Ortloff et al. [67] described the morphology of mature larvae of *Nitidula carnaria* (Schaller 1783); Diaz-Aranda et al. [68] described the morphology of mature larvae of *Nitidula flavomaculata* (Rossi 1790), *O. colon*, and *N. carnaria*; and Williams et al. [69] compared the morphology of adults and mature larvae of *Omosita nearctica* (Kirejtshuk 1987) and *O. colon*. Up to now, the developmental studies on Nitidulidae for the estimation of PMI_min_ are limited to developmental models of *O. colon* (Wang et al. [11]) and preliminary data on the life cycle of *N. carnaria* (Zanetti et al. [70]). There is a lack of studies describing the developmental pattern or morphology of *N. rufipes*.

Considering this, in this study we studied the developmental duration, thermal summation, larval body length, and instar discrimination indexes of *N. rufipes* under seven constant temperatures ranging from 16 to 34 °C. We established several developmental models that could be used to estimate the PMI_min_ using *N. rufipes* in order to provide the lacking basic data for forensic entomology.

## 2. Materials and Methods

### 2.1. Species Identification and Colony Establishment

A field succession study was conducted during the summer (July to August 2021) in Shizuishan City, Ningxia, China (38°98′ N, 106°52′ E) [71]. Upon completion, twenty pairs of *N. rufipes* adults were collected from three pig (*Sus scrofa domestica* L.) carcasses and transported to the Laboratory of Forensic Entomology, Soochow University for further breeding. Species identification was carried out under a Zeiss 2000-C stereomicroscope according to the adult identification key of Zhang et al. [19].

Molecular identification was performed based on the sequenced COI gene of *N. rufipes*, where DNA was extracted for molecular identification from 2 to 5 samples according to the prescribed protocol of the Hipure Insect DNA kit (Guangzhou Magen Biotechnology Co., Ltd., Guangzhou, China). The generated gene sequence has been uploaded to GenBank (accession number: OQ421146). The *N. rufipes* adults were then placed in a 20 × 14 × 8 cm plastic box with 3 cm thick soil and a 3 × 5 cm nylon mesh window on the lid. The colony was placed in a plastic box and reared in a microenvironment incubator set to 25 °C, 70% relative humidity, and an L12: D12 photoperiod for one year. Water was sprayed regularly to maintain humidity. Oats and minced lean pork were mixed with water at a ratio of 1:1:1 as a food source.

### 2.2. Observations of Developmental Duration

After the colony of *N. rufipes* was established, 20 pairs of adults were selected and placed in a Petri dish with a diameter of 12 cm to induce oviposition. Oats and minced lean pork were mixed with water 1:1:1 and were placed on one side of the Petri dish, and the edge of the food was pressed into a slope that fit the bottom of the Petri dish to facilitate adults climbing on and off. A slide (3 × 8 cm) covered with a wet piece of non-woven mask was placed on the other side of the Petri dish to maintain ambient humidity and serve as an oviposition substrate [11].

A total of 20 pairs of adults were reared at each temperature of 16, 19, 22, 25, 28, 31, and 34 °C, and the above steps were repeated to induce oviposition. The oviposition substrate was placed in the Petri dish at 8:00 every day and taken out at 20:00. The number of eggs was counted under the microscope, and the oviposition substrate was placed into an empty Petri dish with a diameter of 12 cm and placed into a microenvironment incubator with the temperature corresponding to their parents. The eggs were sprayed with 1 mL of water every day to prevent desiccation. Observations were made every 4–6 h to record the time of hatching. The hatched individuals were carefully picked out with a soft brush, and the number of hatchlings was calculated by recounting the remaining eggs. The time of hatching was recorded.

The newly hatched larvae were transferred to Petri dishes with a diameter of 4 cm and a height of 1.2 cm in batches of 10. Each Petri dish contained 6 g of minced lean pork, oats, and water in a 1:1:1 mixture provided as a food source. Every 70 hatched larvae were taken as a replicate, and the 7 Petri dishes of each replicate were placed in the same plastic box (20 × 14 × 8 cm) and put in a microenvironment incubator with the same temperature as the respective eggs. The incubators used in this study operated at 70% relative humidity and an L12: D12 photoperiod under the 7 temperatures. A total of 3 replicates or 210 hatched larvae were kept at each temperature.

Larvae crawling away from the food were regarded as entering the post-feeding stage. Dried sandy soil (sand content was about 20%) and distilled water were mixed in a ratio of 7:2 as the pupation substrate and placed into Petri dishes with a diameter of 4 cm. Larvae from the same replicate that entered the post-feeding stage on the same day were transferred to the same pupation dish with tweezers. At most, five larvae were put into each pupation dish. This was repeated until all the larvae entered the post-feeding stage. The time the larvae entered the pupal stage was observed daily under the microscope through the transparent side wall and bottom of the pupation Petri dish. The main distinguishing characteristics of post-feeding larvae and pupae are as follows: Post-feeding larvae that are about to pupate are stationary in a crouched shape, and the segments of the thorax and abdomen are even and slender, similar to feeding larvae, with three pairs of legs visible. The body of the pupal stage is generally straight, shorter, and thicker, with obvious luster, and there is no obvious segmentation of the thorax and abdomen; the three pairs of legs are fixed in front of the thorax together with the elytra; and the abdomen is conical and bifurcated at the end and swings back and forth (Figure 1).

Since Nitidulidae need to pupate in the soil and do not climb out directly like flies after eclosion but go through a dormant period to completely harden their cuticle shell [72], they can only be found by actively digging out the pupal chamber before eclosion to observe the developmental duration of each individual. We recorded the time from post-feeding to eclosion of *N. rufipes* under each temperature through pre-experiments combined with the morphological characteristics of pupal maturity (Figure 1) to ensure that all *N. rufipes* were dug out 1–2 days before eclosion to minimize the possible interference caused by digging. Tweezers were used to carefully scrape the soil away layer by layer to avoid injury or death to the pupae; this was repeated and observed until eclosion of all the individuals.

After eclosion, the body color of *N. rufipes* changes from white to yellow, then to brown, and finally to dark black (Figure 2). The darkening process takes between 2 and 5 days under different temperatures. Due to large individual differences, darkening is considered an unsuitable indicator for the estimation of PMI_min_ in forensic entomology, and the relevant data are not presented in this study.

### 2.3. Determination of Larval Morphological Indexes

The larvae were photographed once a day from the beginning of hatching until more than half of the larvae left the food; 3 Petri dishes were randomly selected in each replicate for each photo shoot, and 3–4 larvae were removed from each Petri dish with a soft brush. The larvae were carefully dipped into the inner wall of the lid of the Petri dish, and clear pictures were taken quickly with a Nikon D700 (Nikon Crop, Tokyo, Japan) digital camera attached to a stereomicroscope (Carl Zeiss, Göttingen, Germany). Larvae were returned to their respective Petri dishes after the shoot. The above photographic process was carried out in 3 replicates to ensure that more than 30 photos were obtained for each developmental day. This is equivalent to randomly sampling 30 individuals as the observation objects of morphological indexes and larval ecdysis events (including first ecdysis and second ecdysis). The three larval morphological indexes of body length, head capsule width, and the distance between the urogomphi were measured using the software ImageJ (https://imagej.net/ (accessed on 26 February 2023)), and different instars were distinguished by observing the morphological characteristics of the larvae in the photos.

### 2.4. Data Analysis

Data analysis was carried out with Origin Pro 8.6 (https://www.originlab.com/ (accessed on 26 February 2023)). Raw data used in this study can be obtained in the Appendix A. Mean duration of the six developmental events and the lower developmental threshold (T_L_) were analyzed using one-way ANOVA + LSD test. The relationship between the number of *N. rufipes* and time in each developmental event from oviposition to eclosion is presented as a histogram. The mean value and standard deviation of each developmental event at each constant temperature were calculated and presented as an isomorphen diagram. The survival rates at different temperatures were analyzed using the modified Kaplan–Meier survival curve, and the scatter plots at each temperature were created according to the relationship between body length and time. Regression analysis equations for estimating age by larval body length were calculated, and an isomegalen diagram was plotted. The thermobiological parameters of *N. rufipes* were evaluated using the linear thermal summation model proposed by Ikemoto and Takai [73] and the curvilinear thermodynamic model (Optim SSI) proposed by Shi et al. [74]. The expression of the SSI model is as follows:$$\frac{\rho_{\emptyset }\;\frac{T}{T_{\emptyset }}\exp\left[\frac{\Delta H_{A}}{R}\;\left(\frac{1}{T_{\emptyset }}-\frac{1}{T}\right)\right]}{1+exp\left[\frac{\Delta H_{L}}{R}\;\left(\frac{1}{T_{L}}-\frac{1}{T}\right)\right]+exp\left[\frac{\Delta H_{H}}{R}\;\left(\frac{1}{T_{H}}-\frac{1}{T}\right)\right]}$$

The value of each parameter was estimated using R 3.5.2 (https://www.r-project.org/ (accessed on 26 February 2023)) using the program developed by Shi et al. [74].

Cluster analysis was performed on the head capsule width and the distance between the urogomphi for larval instar discrimination. The classifier was established by linear discriminant analysis. The classification value was calculated using the formula y = x_1_ × f_1_ + x_2_ × f_2_. In this equation, y represents the classification value of age, x represents the weight of each measured value, and f represents the measured value. The statistical description values of head capsule width and the distance between the urogomphi at each instar, including the mean value and range, are provided.

## 3. Results

### 3.1. Developmental Duration and Isomorphen Diagram

*Nitidula rufipes* showed complete development from egg to adult at all the constant temperatures between 16 and 34 °C. With an increase in temperature, the total developmental duration was shortened from 71.0 ± 4.4 days at 16 °C to 20.8 ± 2.4 days at 34 °C (Table 1). After one-way ANOVA, multiple comparisons between temperatures showed that the duration difference of each developmental event at 16 °C, 19 °C, 22 °C, 25 °C, and 28 °C was statistically significant (*p* < 0.05), while the difference in duration of each developmental event, except for hatching between 31 °C and 34 °C, was not statistically significant (*p* > 0.05). This, combined with the fact that temperatures of 31 °C and 34 °C completed the whole immature stage faster than other temperatures in this study, suggests that the fastest development rate of *N. rufipes* occurs after 28 °C. By observing the changes in the number of individuals (dependent variable) at each developmental event under the seven constant temperatures (Figure 3), the number of individuals showed a right-skewed distribution along with the post-oviposition time (independent variable); thus, the occurrence peak of each developmental event can be seen.

In addition, the standard deviation of developmental events under the same constant temperature in Table 1 gradually increases, suggesting that the number of individuals of different developmental events under the same constant temperature in Figure 3 gradually disperses. The mean value and standard deviation of the time required for the six developmental events of *N. rufipes* under the seven constant temperatures were plotted with post-hatching time as the X axis and temperature as the Y axis on the isomorphen diagram (Figure 4).

### 3.2. Thermal Summation Models and Optim SSI Models

The linear thermal summation models (Figure 5) proposed by Ikemoto and Takai [73] under six developmental events were established with the developmental time as the X axis and the accumulated degree days (ADD) as the Y axis. Using the models, the thermal summation constant and the lower developmental threshold corresponding to each developmental event were obtained (Table 2). During the complete development of *N. rufipes*, the thermal summation constant (K) was 471.40 ± 25.46 degree days, and the lower developmental threshold (T_L_) was 9.65 ± 0.62 °C. There was no statistically significant difference in the lower developmental threshold (T_L_) between first ecdysis, wandering, pupation, and eclosion, but they were statistically different from other developmental events. The R^2^ values of the linear thermal summation models all exceeded 0.95, indicating that the linear fitting of the models was good.

The scatter diagram of each developmental event was plotted with temperature as the X axis and development rate as the Y axis. It can be seen from the scatter diagram that the data points of the highest and lowest temperature in this study, namely, 16 °C and 34 °C, were not linearly related to the 5 intermediate temperatures and belong instead to the nonlinear curve model. The curvilinear thermodynamic Optim SSI model proposed by Shi et al. [74] was used to fit the developmental events of *N. rufipes* under temperatures of 16–34 °C (Figure 6). The results showed that the lower developmental threshold (T_L_), the intrinsic optimum temperature (T_Φ_), and the upper lethal developmental threshold (T_H_) of *N. rufipes* were 10.12, 24.15, and 36.00 °C, respectively (Table 3). The lower developmental threshold (10.12 °C) estimated by the Optim SSI models was similar to the threshold (9.65 °C) obtained by the linear thermal summation models.

### 3.3. Survival Rates and Larval Body Lengths

The survival rate of *N. rufipes* at 16 °C and 34 °C was lower at 40.65% and 43.52%, respectively, while the survival rate of *N. rufipes* at 22 °C and 25 °C was higher at 83.10% and 85.78%, respectively (Figure 7), consistent with the intrinsic optimum temperature of 24.15 °C obtained by the Optim SSI models. The survival rate of the eggs at each constant temperature was very high, except for a small number of eggs that did not hatch at 31 °C and 34 °C. Death rates were highest during the feeding stage at 16 °C and during the feeding, post-feeding, and pupal stages at 34 °C.

The variation in larval body length of *N. rufipes* under the seven constant temperatures is shown in Figure 8. Regression analysis was conducted with larval body length as the dependent variable and the time after hatching as the independent variable, resulting in a simulation equation for the change in larval body length over time (Table 4). The higher the temperature, the faster the increase in larval body length observed.

Using the body length variation curve in Figure 8 and the simulation equation in Table 4, the isomegalen diagram (Figure 9) was constructed with the data of the seven constant-temperature studies. The PMI_min_ can thus be preliminarily estimated in combination with the maximum body length of *N. rufipes* obtained at the case scene.

### 3.4. Larval Instar Discrimination

This section describes the morphological methods developed for the instar discrimination of *N. rufipes* larvae (Figure 10). In addition to the body size differences in the length and width of the *N. rufipes* larvae at different instars, the first instar larvae have a total of 11 dark gray transverse bands in 3 thoracic segments and 8 abdominal segments, among which the first thoracic segment’s transverse band is the thickest and breaks at the midline, while the 2 transverse bands in the second and third thoracic segments are broken on both sides. The head capsule and urogomphi are dark gray and in line with the transverse bands. The transverse bands, head capsule, and urogomphi of the second instar larvae are light orange, and the transverse bands are thicker than those of the first instar larvae. The transverse bands of the first thoracic segment are the thickest and break at the midline. Compared with the first and second instar larvae, the body of the third instar larvae is yellow, and light transmittance is reduced. The transverse band is thicker and nearly square compared to that of the second instar larvae. The color of the transverse band is lighter and almost disappears from the second thoracic segment until the eighth abdominal segment where the color gradually deepens. Compared with the first thoracic segment of the second instar larvae, the first thoracic segment of the third instar larvae has two teardrop-like structures on both sides of the transverse band (Figure 10). To summarize, the shape and color of the transverse bands on the thoracic and abdominal segments, especially the transverse bands on the first thoracic segment, are two of the most important morphological characteristics for the instar discrimination of *N. rufipes* larvae.

In addition to providing morphological characteristics for instar discrimination, a cluster analysis was performed evaluating the two morphological indexes of the head capsule width and the distance between the urogomphi under different temperatures (Figure 11). The data collected at each developmental temperature were divided into three distinct elliptical regions, where each region represented an instar to a total of three instars, consistent with the morphological observation results in Figure 10. The average head capsule width of the first, second, and third instars of *N. rufipes* was 0.21, 0.37, and 0.55 mm, respectively, and the average distance between the urogomphi was 0.11, 0.20, and 0.30 mm, respectively (Table 5). Through the application of linear discriminant curve analysis, the head capsule width and the distance between the urogomphi of *N. rufipes* were used to define the instar discriminant equation as follows: y = 16.00 × f_1_ × 32.25 × f_2_. Table 6 shows that the discrimination accuracy of the first, second, and third instar larvae of *N. rufipes* was 100.00%, 98.21%, and 99.81%, respectively, and the total discrimination accuracy was 99.74%. This suggests that different instars of this species can be identified with accuracy using the head capsule width and the distance between the urogomphi.

## 4. Discussion

A total of 8 case reports and 63 succession studies in forensic entomology involving Nitidulidae were retrieved (Appendix A [2,28,30,31,32,33,34,35,36,37,38,39,40,41,42,43,44,45,46,47,48,49,50,51,52,53,54,55,56,57,58,59,60,61,67,68,69,71,75,76,77,78,79,80,81,82,83,84,85,86,87,88,89,90,91,92,93,94,95,96,97,98,99,100,101,102,103,104,105,106,107]). In terms of succession studies, 32 studies did not identify species (14 studies (43.75%) from 1981 to 2010 and 18 studies (56.25%) from 2010 to 2023), and 31 studies did identify species (8 studies (25.81%) from 1981 to 2010 and 23 studies (74.19%) from 2010 to 2023). The literature survey revealed that the attention to and identification of Nitidulidae by forensic personnel has improved significantly in recent years. Among the 31 succession studies that identified Nitidulidae species, 13 species were considered to be of forensic significance and were mentioned a total of 78 times, with an average of 2.52 species per study. This suggests that most species of necrophagous or saprophagous Nitidulidae are widely distributed and prone to co-occurrence. In addition, this family was more likely to appear in the active decay stage through to the remains stage, was more prevalent in forest and grassland succession studies, and was less prevalent in urban areas and indoors (Appendix A). *Omosita* Erichson, 1843 and *Nitidula* Fabricius, 1775 were the two genera most associated with carcasses and appeared more frequently in studies conducted from April to August. In this study, a large number of *N. rufipes* were found on domestic pig carcasses placed in July, consistent with the results of previous studies.

Most of the relevant literature details the morphological description of mature larvae or adults of *N. rufipes* but lacks the description of their first and second instar larvae. Frątczak and Matuszewski argued that [108], at present, the only way to discriminate the larval instars of most forensic important beetles is to measure their size since there is a lack of morphological characteristics for specific instars. According to the developmental duration of this study (Table 1), the duration of the first and second instars of the larvae was very close to that of the third instar, so their significance in the PMI_min_ estimation of forensic entomology should be equally important. The morphological description and cluster analysis of the three larval instars in this study were conducive to the discrimination of the instars of *N. rufipes* larvae. It was also helpful to distinguish the larvae from other species of Nitidulidae and necrophagous beetles on carcasses, thus providing valuable information for forensic entomology studies and case investigations [67,68].

The pupal stage of the beetle also accounts for a large part of the immature stage, but, compared with the obvious morphological changes in the pupa of flies [109,110], the morphological changes of most beetles, including *N. rufipes*, are very slight. The study of Novak et al. [111] on the compound eyes of *Necrodes littoralis* (Linnaeus, 1758) helps to improve the ability of Silphidae evidence to estimate the PMI_min_. In this study, the color changes of the compound eyes and mouthparts can be used as two important indicators for the preliminary estimation of the pupal maturity of *N. rufipes*. As shown in Figure 1, the compound eyes of *N. rufipes* were almost invisible on the first day after pupation and were completely colored on the fourth day. The tip of the mouthparts began to color on the fifth day, and the color gradually deepened until the seventh day. Therefore, an uncolored compound eye indicates that the pupation of *N. rufipes* is close, and the coloring of mouthparts is a sign that the pupa of *N. rufipes* is relatively mature. In addition, the gradual shortening and widening of the pupa of *N. rufipes* is also key in the development of the pupal stage, mainly reflected in the shortening of the abdomen. Future studies on gene expression in the pupal stage can provide more accurate and diversified molecular markers for PMI_min_ estimation. Andersen et al. [72] argued that the hardening and coloring of the cuticle after eclosion was due to a series of enzyme reactions that occur simultaneously in the insect’s body. We tried to increase the maximum length of the PMI_min_ estimation by observing the time of darkening after the eclosion of each individual, and the results showed that the time varies greatly among individuals, ranging from 2 to 5 days under the same temperature. There was no significant temperature-dependent darkening observed after eclosion, so this developmental event may not be suitable as a time point indicator for the PMI_min_ estimation in forensic entomology.

Many succession studies have mentioned that Dermestidae and Nitidulidae are two very important families of necrophagous beetles in the late stage of body decomposition [97,112,113,114]. Even though they often appear on carcasses at the same time, the former tends to choose drier environments, while the latter tends to be saprophagous, preferring moist rotten tissue or soil soaked in rotten liquid [11,85,115]. In this study, the oviposition substrate was regularly sprayed with water, and the food and pupation substrate were kept moist as per Nitidulidae preference, which ensured the repeatability and reproducibility of the study. Whether the water content of the food has an impact on the larval development rate needs further study. The water content of the pupation substrate was a relatively suitable proportion obtained through pre-experimental data which showed that if the soil humidity was low, loose soil particles could not be built into a pupal chamber, and if the humidity was too high, the larvae would drill many tunnels in the soil but could not form a closed pupal chamber, which might hinder their oxygen acquisition after being closed. Both of the above two conditions would lead to a delay in pupation or even the inability to pupate, leading to the death of the post-feeding larvae.

The adults and larvae of Nitidulidae are small and difficult to study [11], but the distance of pupation from the carcass should be much closer than that of other beetles or Diptera. If Nitidulidae have colonized the carcass, they should be present in the soil soaked by the liquid of body decomposition [85]. This makes it particularly important to study their post-feeding and pupal stages. After crawling away from the food and being tweezed onto the pupation substrate, most of the larvae in this study drilled directly into the soil to prepare for pupation. However, even with similar soil humidity, the larvae in the post-feeding stage at 34 °C crawled out of the pupation substrate much more frequently than at other temperatures and were very restless even when they were put back onto the pupation substrate. This indicates that even if the larvae grow faster at higher temperatures during the feeding stage, very high temperatures are not suitable for their pupation. In addition, the high mortality rate, which is possibly due to faster dehydration at 34 °C compared with other temperatures, supports this finding. The larvae did not seem to adapt well to the dry environment of the plastic box after leaving the wet food in the post-feeding stage; thus, spraying water on the inner wall of the plastic box in the post-feeding stage and appropriately increasing the frequency of transferring the larvae to the pupation substrate to 2–3 times a day could largely avoid the death of larvae in the post-feeding stage. In addition to species differences, this also explains the lower mortality rate found in this study compared to that of Wang et al. [11] on the development of *O. colon*.

Wang et al. [11] found that the survival rate of *O. colon* was highest at 25 °C, while the survival rate at 16 °C was the lowest, and the mortality rate of the pupal stage was low, which is consistent with the findings of this study, but *O. colon* failed to complete its development at 34 °C. In this study, although the mortality rate of *N. rufipes* was relatively high at 34 °C, some individuals still successfully completed development, suggesting that *N. rufipes* is more resistant to high temperatures than *O. colon*. In terms of the developmental duration, the total time from oviposition to eclosion of *N. rufipes* is slightly shorter than that of *O. colon* at 16–31 °C. For example, *N. rufipes* needs 71.05 ± 4.37 days to eclose at 16 °C, while *O. colon* needs 95.3 ± 11.4 days to eclose. At 31 °C, it took 20.95 ± 2.31 days for *N. rufipes* to eclose and 25.2 ± 2.6 days for *O. colon* to eclose. Compared with some developmental studies of non-necrophagous Nitidulidae species, in this study, the lower developmental threshold of *N. rufipes* was 9.65 °C, lower than that of *Carpophilus hemipterus* (Linnaeus, 1758) (14.6 °C), *Carpophilus mutilatus* Erichson 1843 (15.3 °C), and *Carpophilus humeralis* (Fabricius 1798) (15.4 °C) [116], but higher than that of *Meligethes aeneus* (Fabricius, 1775) (4.0 °C) [117]. *Glischrochilus quadrisignatus* (Say, 1835) [118] could not complete its entire life cycle at 10.0 °C, which is similar to what we observed with *N. rufipes*. In addition, *C. hemipterus*, *C. utilates*, *C. humeralis*, and *G. quadrisignatus* can complete their entire life cycle at the maximum temperatures of 40.0 °C, 37.5 °C, 32.5 °C, and 30.0 °C, respectively, where the upper lethal developmental threshold of *N. rufipes* was estimated to be 36.00 °C. Therefore, different species of Nitidulidae have different tolerances to high and low temperatures during development. The specific characteristics of these species may determine their ecological niche to a certain extent, which can be seen in Appendix A for some necrophagous Nitidulidae species.

Since most necrophagous beetles, including the Nitidulidae, have a lower fecundity and a longer developmental time than flies, the continuous killing and sampling of larvae are destructive [119,120,121]. Moreover, the larvae of Nitidulidae are small, and their head capsules and urogomphi are too small to be measured with calipers, making them suitable for in vivo measurement [122]. Frątczak and Matuszewski [123] found that the temperature change when moving insects from the incubator to room temperature and the handling pressure on the larvae during the measurement affected the developmental duration of *Creophilus maxillosus* (Linnaeus, 1758), rendering the measurements in vivo slightly longer and the size of the adults smaller than that of the individuals not measured, thus affecting the quality of the developmental data obtained. In this study, 210 individuals were placed at each temperature, of which only about 30 individuals were measured on each developmental day during the feeding stage. By increasing the number of individuals, the average measurement-derived stress and temperature change influence was minimized, and the quality of the data was improved.

It must be noted that the differential effects of fluctuating temperature and constant temperature on insect development rate will limit the application of the data in this study to a certain extent. Based on separate studies of two Hymenoptera species, *Tamarixia radiata* (Waterson, 1922) and *Diaphorencyrtus aligarhensis* (Shafee, Alam, and Argarwal 1975), McCalla et al. [124] and Milosavljević et al. [125] found that they completed their development faster and survived longer at cooler fluctuating temperatures (15 °C) than at the corresponding mean constant temperatures. These species also completed their development slower and survived for shorter periods at higher fluctuating temperatures (35 °C) than at constant temperatures. The lower developmental thresholds of *D. aligarhensis* estimated by the linear and nonlinear equations at fluctuating temperatures were higher than those under a constant temperature regimen. Similarly, Hagstrum and Milliken [126] noted that the developmental times of the red flour beetle, *Tribolium castaneum* (Herbst, 1797), were shorter at constant temperatures above 25–30 °C than that at fluctuating temperatures, while the opposite was true at lower temperatures. Therefore, the impact of insect development rate on the PMI_min_ estimation needs to be considered carefully [5]. It is necessary to reasonably validate the constant temperature data obtained under laboratory standard conditions with the data obtained under the practical fluctuating temperature conditions. This will drastically improve the estimation of the PMI_min_ in forensic entomology, and it is a key problem that researchers need to focus on in future studies.

## 5. Conclusions

As a necrophagous beetle with great value and potential in forensic entomology, *N. rufipes* can complete its development at temperatures between 16 and 34 °C, where developmental time decreased with increasing temperature. In this study, the linear thermal summation models and Optim SSI models were established to understand the thermobiological characteristics of *N. rufipes*. The isomorphen diagram, isomegalen diagram, and equations of larval body length changes with time after hatching were established, and the cluster analysis of the head capsule width and the distance between the urogomphi was combined with morphological characteristics to aid in the instar discrimination, thus providing relatively complete developmental data for *N. rufipes*. The developmental data allow the estimation of the age of the immature stages that ultimately facilitates the estimation of PMI_min_.

## Figures and Tables

**Figure 1 insects-14-00299-f001:**
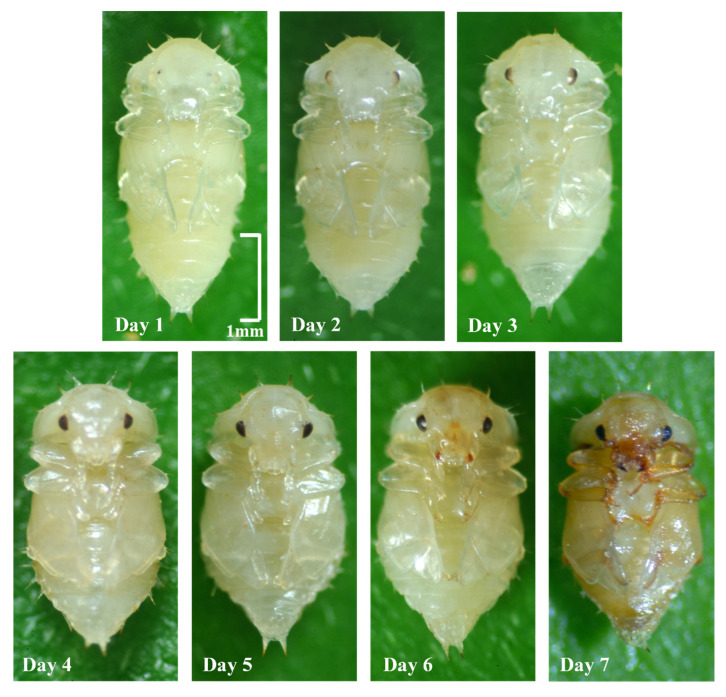
Optical microscope images of daily developmental changes in *Nitidula rufipes* after pupation.

**Figure 2 insects-14-00299-f002:**
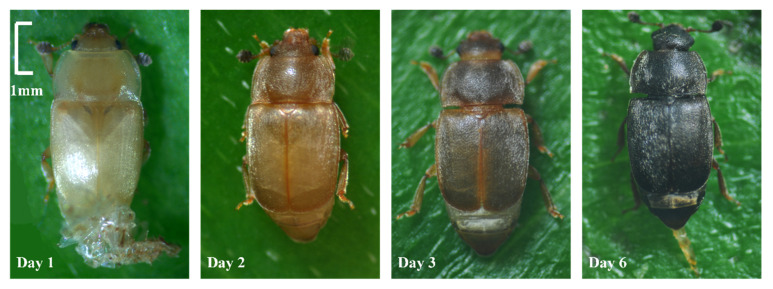
Optical microscope images of the darkening process of *Nitidula rufipes* after eclosion.

**Figure 3 insects-14-00299-f003:**
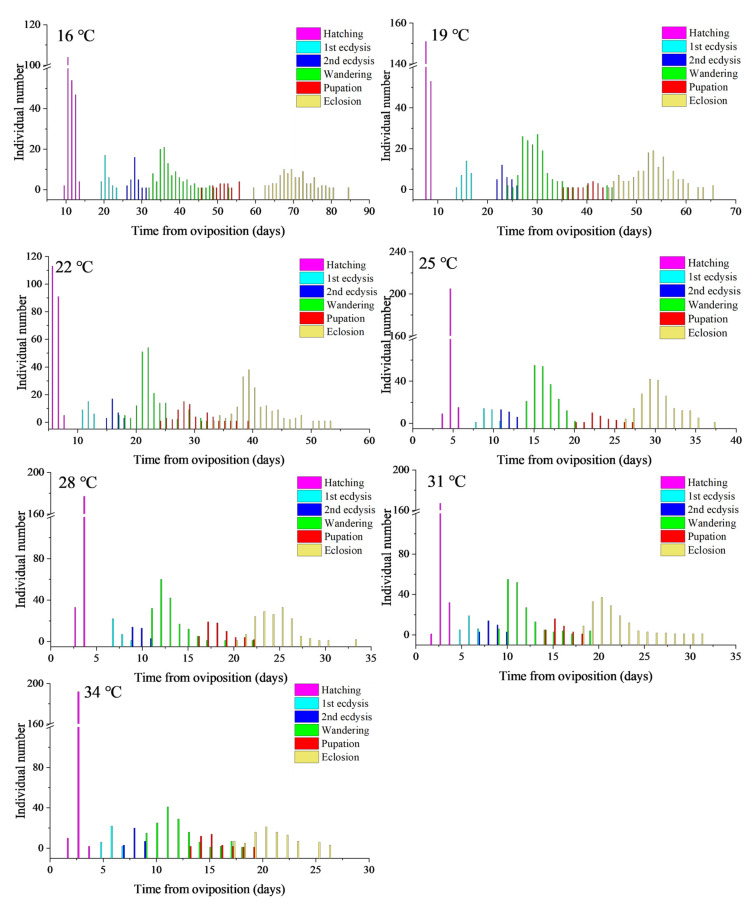
The relationship between the developmental events and individual numbers of *Nitidula rufipes* under seven constant temperatures with humidity of 70% and a photoperiod of 12:12 h L: D.

**Figure 4 insects-14-00299-f004:**
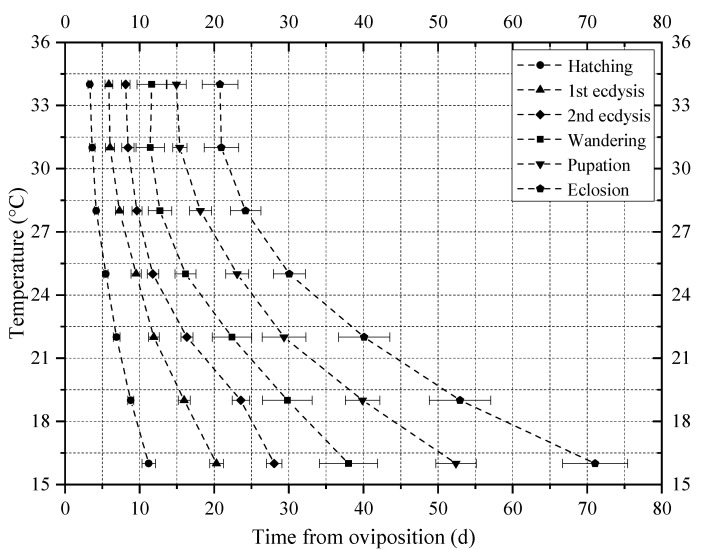
Isomorphen diagram of *Nitidula rufipes*. The relationship between seven constant temperatures and the duration in days from oviposition to each of the six developmental events was plotted. Each curve corresponds to a particular developmental event. The error bar represents the SD.

**Figure 5 insects-14-00299-f005:**
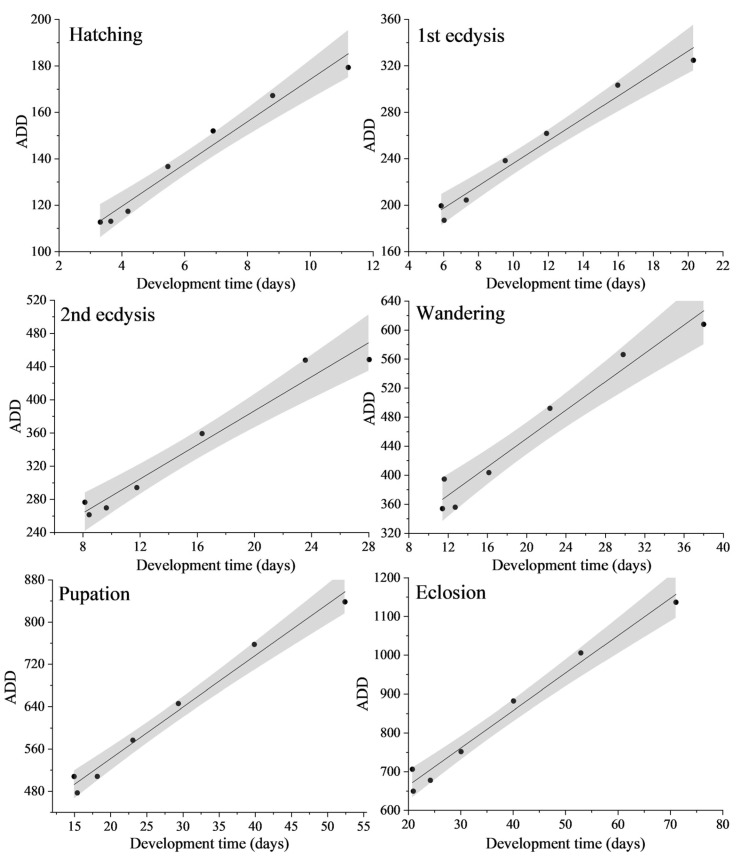
Thermal summation models of the six developmental stages of *Nitidula rufipes*. ● Represents the mean value. The solid line represents the regression line. The gray area represents the 95% confidence interval.

**Figure 6 insects-14-00299-f006:**
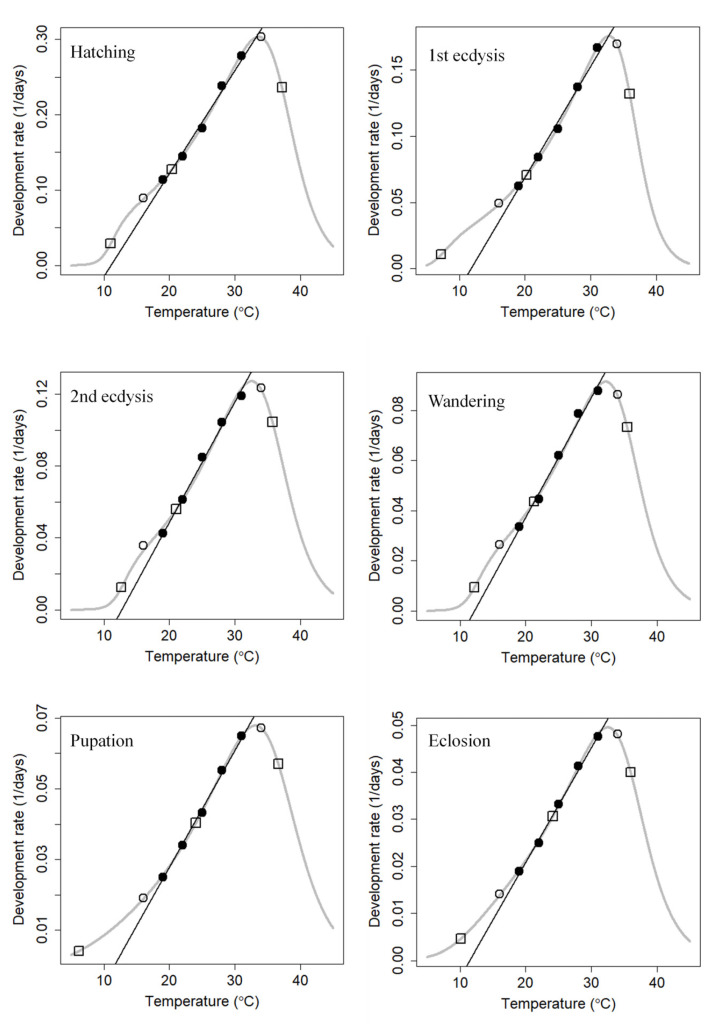
Curvilinear thermodynamic Optim SSI models of the six developmental stages of *Nitidula rufipes*. The curved line indicates the development rate predicted by the Optim SSI model of Shi et al. The three open squares represent the predicted mean development rates at T_L_, T_Φ_, and T_H_. The black circles represent the data used for the linear fitting by the reduced major axis, whereas the white circles are the data excluded from the linear fitting.

**Figure 7 insects-14-00299-f007:**
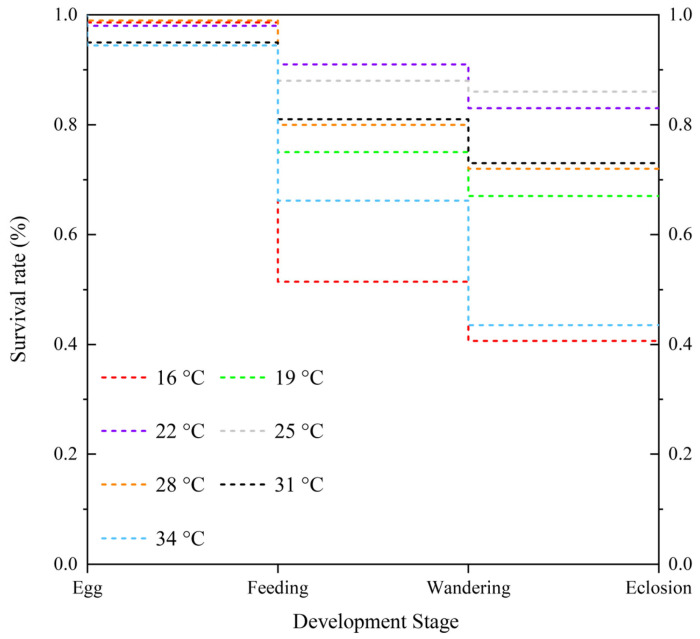
Modified Kaplan–Meier survival curves of *Nitidula rufipes* under seven constant temperatures.

**Figure 8 insects-14-00299-f008:**
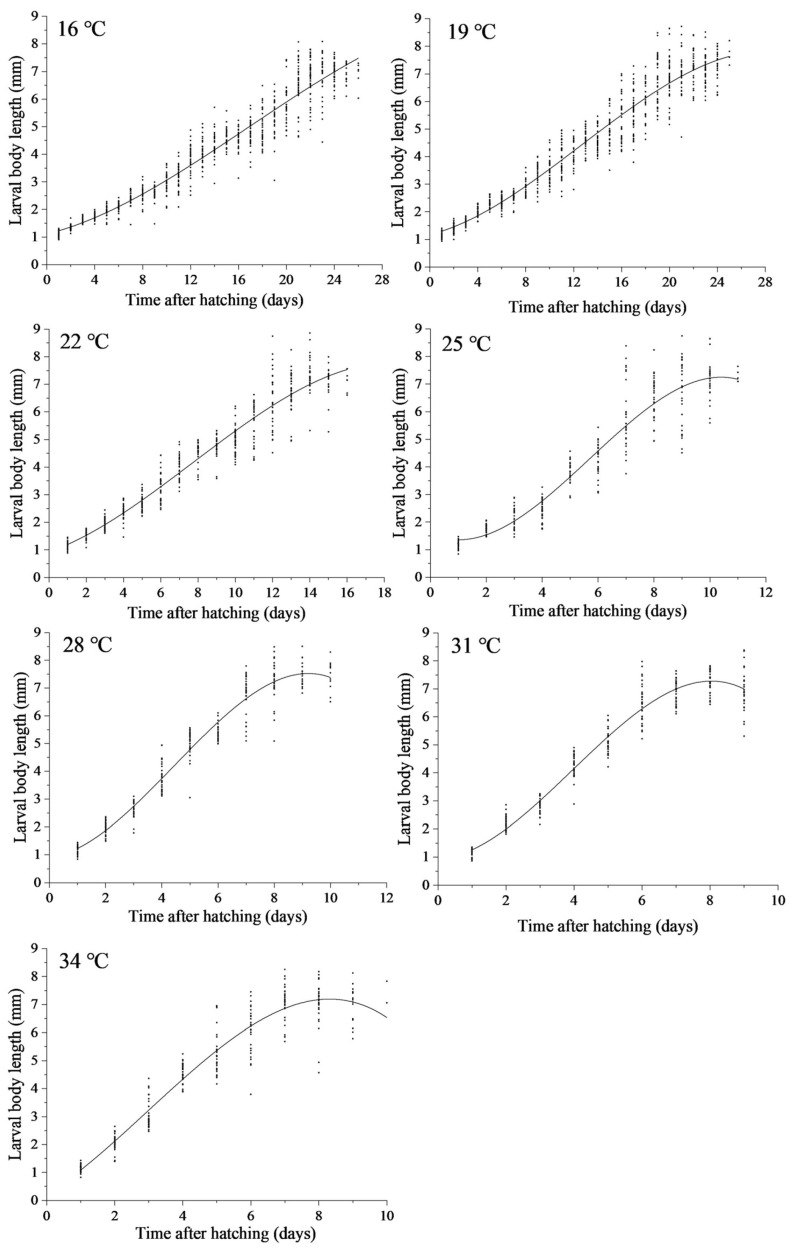
The larval body length of *Nitidula rufipes* changes over time after hatching under seven constant temperatures.

**Figure 9 insects-14-00299-f009:**
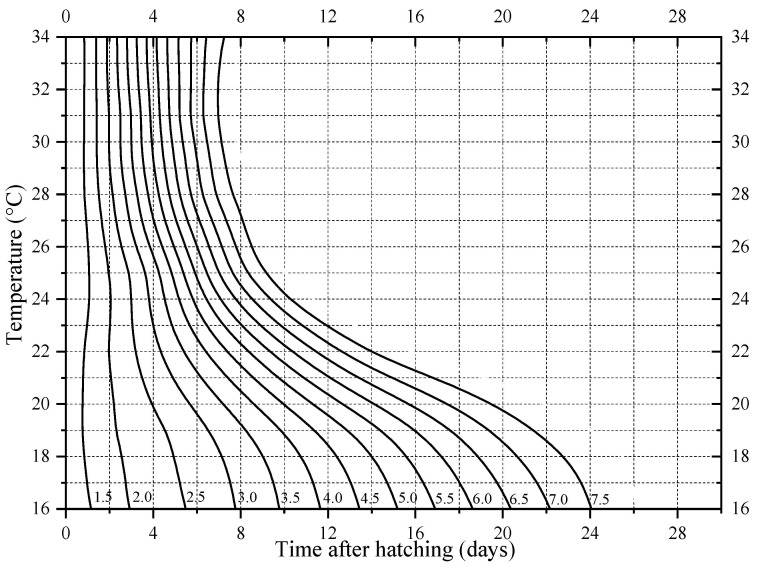
Isomegalen diagram of *Nitidula rufipes*. Each contour line represents larval body length (mm) (Z axis) related to age (X axis) and temperature during development (Y axis).

**Figure 10 insects-14-00299-f010:**
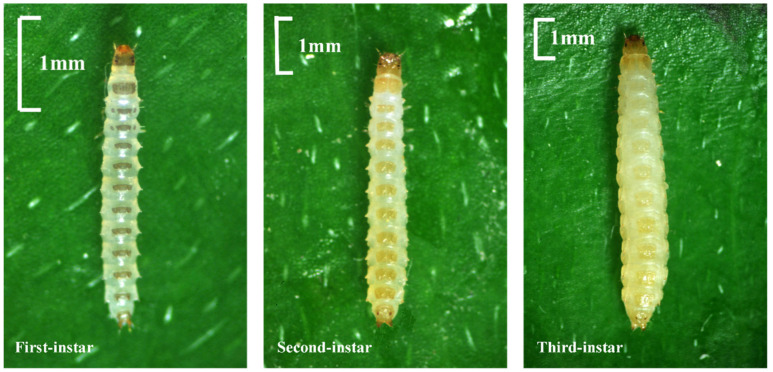
Optical microscope images of the first, second, and third instar larvae of *Nitidula rufipes*.

**Figure 11 insects-14-00299-f011:**
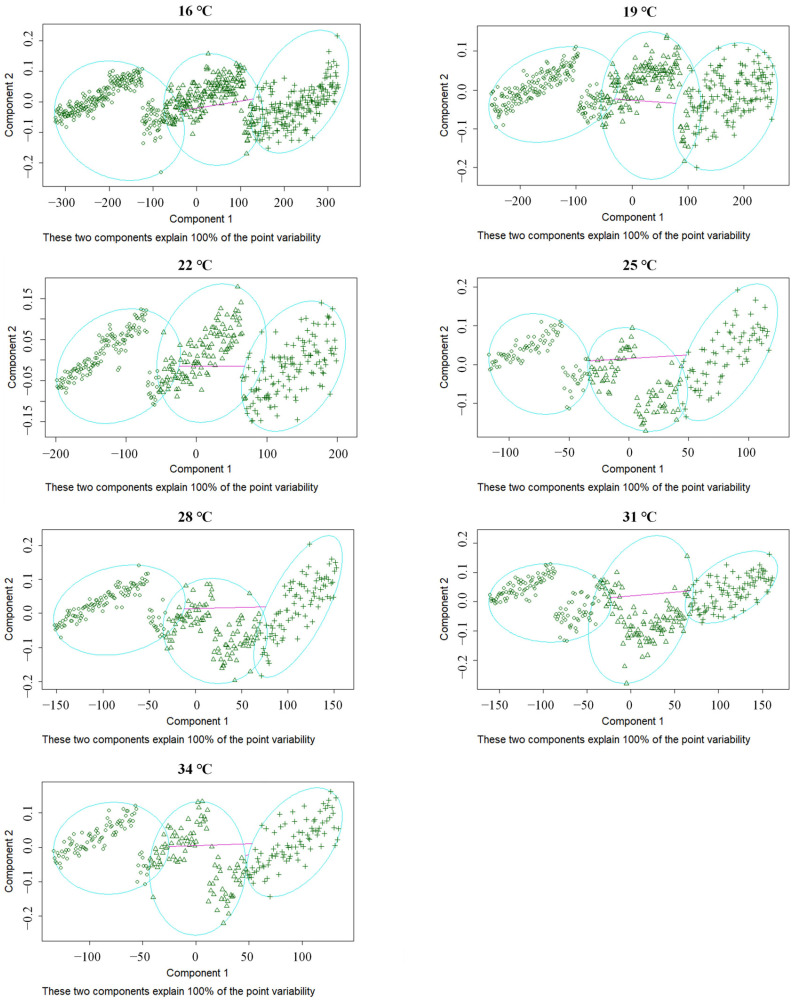
Cluster analysis diagram of the head capsule width and the distance between the urogomphi of *Nitidula rufipes* larvae at three instars under seven constant temperatures. Each blue oval represents measurements from one instar.

**Table 1 insects-14-00299-t001:** Mean duration (days ± SD) of the six developmental events of *Nitidula rufipes* under seven constant temperatures with humidity of 70% and a photoperiod of 12:12 h L: D.

Developmental Events		Temperature (°C)						
		16 °C	19 °C	22 °C	25 °C	28 °C	31 °C	34 °C
Hatching	min	10.14 ± 0.38 ^a^	8.44 ± 0.42 ^b^	6.14 ± 0.17 ^c^	4.79 ± 0.21 ^d^	3.71 ± 0.15 ^e^	3.10 ± 0.26 ^f^	2.82 ± 0.07 ^f^
	mean	11.21 ± 0.89 ^a^	8.80 ± 0.40 ^b^	6.91 ± 0.44 ^c^	5.47 ± 0.28 ^d^	4.19 ± 0.26 ^e^	3.65 ± 0.36 ^f^	3.32 ± 0.25 ^g^
	max	12.14 ± 1.06 ^a^	9.65 ± 0.20 ^b^	7.97 ± 0.48 ^c^	6.13 ± 0.05 ^d^	4.64 ± 0.17 ^e^	4.31 ± 0.12 ^e^	3.83 ± 0.30 ^e^
1st ecdysis	min	18.47 ± 0.51 ^a^	14.44 ± 0.42 ^b^	10.47 ± 0.75 ^c^	8.46 ± 0.77 ^d^	6.71 ± 0.15 ^e^	5.77 ± 0.79 ^ef^	5.15 ± 0.59 ^f^
	mean	20.30 ± 0.92 ^a^	16.00 ± 0.81 ^b^	11.90 ± 0.71 ^c^	9.53 ± 0.68 ^d^	7.30 ± 0.53 ^e^	6.03 ± 0.61 ^f^	5.87 ± 0.51 ^f^
	max	22.47 ± 0.85 ^a^	16.99 ± 0.55 ^b^	12.64 ± 0.17 ^c^	10.79 ± 0.54 ^d^	7.64 ± 0.17 ^e^	6.64 ± 0.64 ^ef^	6.17 ± 0.85 ^f^
2nd ecdysis	min	26.14 ± 1.00 ^a^	21.11 ± 0.67 ^b^	15.14 ± 0.17 ^c^	11.46 ± 0.41 ^d^	8.71 ± 0.15 ^e^	7.77 ± 0.79 ^f^	6.82 ± 0.07 ^g^
	mean	28.03 ± 1.03 ^a^	23.57 ± 1.17 ^b^	16.33 ± 0.80 ^c^	11.77 ± 0.77 ^d^	9.63 ± 0.67 ^e^	8.43 ± 0.82 ^f^	8.13 ± 0.57 ^f^
	max	30.47 ± 0.48 ^a^	25.32 ± 0.38 ^b^	16.97 ± 0.48 ^c^	13.13 ± 0.05 ^d^	9.97 ± 0.48 ^e^	9.31 ± 0.12 ^f^	8.83 ± 0.30 ^f^
Wandering	min	33.14 ± 1.58 ^a^	25.11 ± 0.98 ^b^	19.47 ± 2.17 ^c^	14.12 ± 0.77 ^d^	11.04 ± 0.73 ^e^	9.43 ± 0.82 ^ef^	8.82 ± 0.07 ^f^
	mean	38.00 ± 3.91 ^a^	29.80 ± 3.33 ^b^	22.37 ± 2.64 ^c^	16.15 ± 1.41 ^d^	12.71 ± 1.57 ^e^	11.42 ± 1.91 ^f^	11.61 ± 1.96 ^f^
	max	44.14 ± 7.47 ^a^	38.32 ± 4.56 ^a^	27.64 ± 4.58 ^b^	19.46 ± 1.20 ^c^	18.97 ± 2.69 ^c^	17.64 ± 1.56 ^c^	17.17 ± 0.36 ^c^
Pupation	min	48.47 ± 2.50 ^a^	36.44 ± 1.42 ^b^	26.14 ± 1.79 ^c^	20.79 ± 1.21 ^d^	16.38 ± 0.63 ^e^	14.77 ± 0.79 ^ef^	13.50 ± 0.85 ^f^
	mean	52.40 ± 2.70 ^a^	39.88 ± 2.31 ^b^	29.36 ± 2.94 ^c^	23.07 ± 1.54 ^d^	18.15 ± 1.47 ^e^	15.38 ± 0.95 ^f^	14.94 ± 1.28 ^f^
	max	55.14 ± 1.02 ^a^	45.32 ± 2.36 ^b^	34.64 ± 3.53 ^c^	26.13 ± 1.05 ^d^	20.97 ± 0.48 ^e^	17.98 ± 1.22 ^e^	17.50 ± 1.82 ^e^
Eclosion	min	65.14 ± 5.01 ^a^	45.78 ± 1.85 ^b^	35.8 ± 2.36 ^c^	26.46 ± 1.34 ^d^	20.71 ± 1.15 ^de^	18.77 ± 0.79 ^de^	17.17 ± 0.85 ^e^
	mean	71.05 ± 4.37 ^a^	52.94 ± 4.12 ^b^	40.09 ± 3.43 ^c^	30.07 ± 2.13 ^d^	24.20 ± 2.04 ^e^	20.95 ± 2.31 ^f^	20.78 ± 2.40 ^f^
	max	80.47 ± 4.29 ^a^	59.65 ± 5.15 ^b^	48.3 ± 5.13 ^c^	35.46 ± 1.48 ^d^	30.3 ± 1.94 ^de^	30.31 ± 1.01 ^de^	26.5 ± 0.84 ^e^

Values in the same row followed by the same letter do not differ significantly from each other based on a one-way ANOVA + LSD test at *p* < 0.05.

**Table 2 insects-14-00299-t002:** Lower developmental thresholds (T_L_) and thermal summation constants (K) for the six developmental stages of *Nitidula rufipes*.

Developmental Stage	K ± SE (Degree Days)	T_L_ ± SE (°C)	R^2^
Hatching	83.12 ± 4.73	9.11 ± 0.70 ^a^	0.97
1st ecdysis	139.39 ± 8.74	9.67 ± 0.72 ^b^	0.97
2nd ecdysis	181.50 ± 14.96	10.26 ± 0.89 ^c^	0.96
Wandering	255.15 ± 20.08	9.78 ± 0.90 ^b^	0.95
Pupation	347.21 ± 17.41	9.74 ± 0.57 ^b^	0.98
Eclosion	471.40 ± 25.46	9.65 ± 0.62 ^b^	0.98

Values in the same column followed by the same letter do not differ significantly from each other based on a one-way ANOVA + LSD test at *p* < 0.05.

**Table 3 insects-14-00299-t003:** Parameter estimates for the six developmental stages of *Nitidula rufipes* according to the curvilinear thermodynamic Optim SSI models.

Parameter (Unit)	Hatching	1st Ecdysis	2nd Ecdysis	Wandering	Pupation	Emergence
*T_Φ_* (°C)	20.38	20.23	21.06	21.30	24.05	24.15
*ρ_Φ_* (day^−1^)	0.13	0.07	0.06	0.04	0.04	0.03
*∆H_A_* (cal/mol)	1.35 × 10^4^	1.45 × 10^4^	1.56 × 10^4^	1.49 × 10^4^	1.42 × 10^4^	1.40 × 10^4^
*∆H_L_* (cal/mol)	−1.53 × 10^5^	−1.19 × 10^5^	−1.44 × 10^5^	−1.31 × 10^5^	−3.87 × 10^4^	−5.72 × 10^4^
*∆H_H_* (cal/mol)	8.67 × 10^4^	1.07 × 10^5^	8.19 × 10^4^	8.56 × 10^4^	6.92 × 10^4^	7.88 × 10^4^
*T_L_* (°C)	11.03	7.24	12.68	12.19	6.20	10.12
*T_H_* (°C)	37.19	35.96	35.76	35.46	36.64	36.00
χ^2^	2.73 × 10^−4^	7.72 × 10^−5^	9.62 × 10^−4^	3.67 × 10^−4^	2.51 × 10^−5^	4.06 × 10^−5^
R^2^	0.999	0.999	0.992	0.994	0.999	0.999

**Table 4 insects-14-00299-t004:** Equations, degrees of freedom (df), and coefficient of determination (R^2^) of the relationship between the time after hatching (T) (day) and the body length of *Nitidula rufipes* larvae (L) (mm) under seven constant temperatures.

Temperature (°C)	Equation	df	R^2^
16	L = −2.146E-4T^3^ + 0.011T^2^ + 0.109T + 1.107	793	0.929
19	L = −4.651E-4T^3^ + 0.018T^2^ + 0.108T + 1.175	799	0.926
22	L = −0.002T^3^ + 0.036T^2^ + 0.235T + 0.927	502	0.931
25	L = −0.014T^3^ + 0.245T^2^ − 0.455T + 1.584	311	0.897
28	L = −0.015T^3^ + 0.204T^2^ + 0.140T + 0.900	276	0.952
31	L = −0.023T^3^ + 0.278T^2^ + 0.067T + 0.938	300	0.955
34	L = −0.013T^3^ + 0.117T^2^ + 0.773T + 0.211	265	0.922

**Table 5 insects-14-00299-t005:** Means and ranges of the head capsule width and the distance of urogomphi (mm) at each instar.

Morphological Indexes	Instar	Mean ± SD	Range	Sample Size
The widths of head capsules	1st	0.21 ± 0.02	0.15–0.28	1055
	2nd	0.37 ± 0.04	0.24–0.49	905
	3rd	0.55 ± 0.05	0.37–0.73	1387
The distance of urogomphi	1st	0.11 ± 0.01	0.07–0.16	1055
	2nd	0.20 ± 0.02	0.14–0.28	905
	3rd	0.30 ± 0.03	0.20–0.39	1387

**Table 6 insects-14-00299-t006:** Classification matrix of the three instars of the measured larvae of *Nitidula rufipes*.

Instar	Sample Size	Size Prediction of Classification Precision			Precision Rate
		1st	2nd	3rd	
1st	802	802	0	0	100.00%
2nd	838	0	823	15	98.21%
3rd	1030	0	2	1028	99.81%
Total	2660	802	825	1043	99.74%

## Data Availability

The data presented in this study are available in the Appendix A.

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
