# Peer review of "Temperature-Dependent Development of *Nitidula rufipes* (Linnaeus, 1767) (Coleoptera: Nitidulidae) and Its Significance in Estimating Minimum Postmortem Interval"

_insects, 2023, doi:10.3390/insects14030299_

Round 1

Reviewer 1 Report

This paper is an extensive study of development rates and immature morphology of a very understudied but forensically important beetle. It provides useful developmental data, thresholds, ADD and morphological changes which will be very valuable in forensic entomology. Images and figures are well done. The data appear to be well analyzed and illustrated.

My main concern is why were mean developmental times presented when almost all similar data are presented as minimums which Are usually used in PMImin calculations. A range would have been more helpful.

I presume you mean Dermestidae at line 428?

Reviewer 2 Report

The authors have graphed and presented their results clearly, drawing some attention to the implications of their findings. I found the study of interest and a good contribution to the knowledge of bio ecology of sap beetles. The methods used are appropriate for the objectives of the work and, in general, well depicted. The resulting figures are sufficient, informative, and of good quality helping to follow the reasoning throughout the manuscript. The discussion of results and comments on future research should be improved if the paper is to be accepted for publication in Insects.

My primary concern is that the authors are extrapolating the applicability of their results beyond what the design supports. These are only data from a set of seven highly artificial constant laboratory conditions, so the inference power of the paper is very limited, but authors do not acknowledge this detail at all and need to be more forthcoming. The effect of fluctuating temperature profiles on sap beetles was not investigated in this study. This is a critical limitation of the study, and the authors must concede and discuss this. The interaction of cyclic temperatures with nonlinear characteristics of sap beetle development curves can introduce significant deviations from the results obtained here, and especially at the lower and higher temperatures of development functions. Studies across a broader set of fluctuating temperature regimes are therefore encouraged so that more realistic effect of temperature on biological parameters of sap beetles could be elucidated, as this is the closest to temperature fluctuations that occur in the field. So, I am suggesting to the authors to tone-down the language a little and admit that there are still substantive uncertainties to be considered.

Some of the authors’ statements would be much stronger if they tie their work to the body of literature that has built up on the bio ecology of hymenopterans, e.g., Journal of Economic Entomology 112:1560-1574 and Journal of Economic Entomology 112:1062-1072. These studies provide strong evidence that daily temperature fluctuations significantly affected development times and longevity of hymenopterans studied, resulting in marked deviations and potentially erroneous predictions when compared to their constant temperature regimen counterparts. In these particular studies, each fluctuating temperature profile was modeled after field recorded temperatures that had the desired average target temperature. These are the first studies ever to undergo such analysis. This article should provide details on all these fronts to provide the proper context for the work. This is not to diminish the data gathered in this study, as they are of value. But it is important for the authors not to overgeneralize, and to warn the reader, including regulatory agencies, against doing so as well. Adding these details will improve the discussion.

Good luck!

Reviewer 3 Report

The authors investigated the influence of temperature on the rate of pre-adult development of a sap beetle Nitidula rufipes with the aim to use this insect in forensic entomology. In addition, morphology of larval and pupal stages was described. The experiments were well designed and conducted. The statistical analysis is correct. The text is mostly well and clearly written. The results of the study are certainly important for both fundamental and applied (forensic) entomology and therefore the manuscript can be published, although it certainly needs corrections and improvements (see below).

Lines 107-108: If the microenvironment incubator was set to 70% relative humidity, it can be expected that this parameter was regulated automatically. Then why it was necessary to regularly spray water to maintain humidity? On the other hand, if you regularly sprayed water, the relative air humidity can be higher than the incubator set value. Was the relative air humidity measured during the experiment? Please, explain it in more details because (as noted in the Discussion) humidity can have a significant impact on larval mortality and rate of development.

Line 114: Considering that Petri is a proper name of a person, this word must begin with uppercase letter here and everywhere below.

Lines 120-121: Please, explain why just these lowest (16) and highest (34) temperatures were selected. Was this choice based on earlier studies or on pilot tests or on something else? In particular, why temperatures above 34 C were not tested? This would allow direct determination (not only indirect estimation with the model) of the higher developmental threshold.

Lines 124-125: Was the relative air humidity regulated in these thermostats? What about photoperiod during the embryo and larval development? I have found the replies much lower, in Table 1 (lines 237-238), but this important information should be also clearly indicated in the Methods section.

Line 137: not “was regarded as them entering” but “were regarded as entering”.

Line 138: Was it sand or namely sandy soil? If the second is true, what was the approximated sand / clay proportion?

Line 168: Please, explain why you use the term ‘discoloration’ which is commonly used for the color change to colorless (white). In your case, just the opposite was observed (I would call it darkening).

Line 223: First, “similar” is not a suitable word in this particular case. The difference can be either statistically significant (at the given sample size) or not. Second, as clearly seen from Table 1, the duration of most developmental stages (excluding wandering) at 31 °C was somewhat longer than that at 34 °C. Therefore I would not agree that the fastest development can be observed at some temperature between 31 and 34 °C.

Table 1: Significance of the difference between temperatures should be calculated for each parameter by the Tukey HSD (or some other test for multiple pairwise comparisons) and indicated in a common way (e.g. by letters within each column). In particular, the significance of the difference between the data for 31 and 34 °C is interesting (see my comments to line 223).

Figures 3, 5, 8, and 11: I would strongly suggest increasing font size used for the legends along the axes. The font size used for Figs. 6 and 7 is fine.

Table 2: Was the difference between the lower developmental thresholds calculated for different developmental stages statistically significant? (see my comments to Table 1).

Finally, I would suggest including in the Discussion the comparison of your data on the lower developmental threshold and thermal summation constant with the results of earlier studies on thermolability of non-necrophagous Nitidulidae species development, for example:

Mussen E.C., Chiang H.C.; 1974; Development of the picnic beetle, Glischrochilus quadrisignatus (Say), at various temperatures.; Environmental Entomology; 3; 1032-1034

Nielsen, P. S. and J. Axelsen; 1988; Developmental time and mortality of the immature stages of the pollen beetle (Meligethes aeneus F.) under natural conditions.; Journal of Applied Entomology; 105; 198-204

James, D. G. and B. Vogele; 2000; Development and survivorship of Carpophilus hemipterus (L.), Carpophilus mutilatus Erichson and Carpophilus humeralis (F.) (Coleoptera: Nitidulidae) over a range of constant temperatures; Australian Journal of Entomology; 39;  180-184

Round 2

Reviewer 2 Report

Authors have done fine job addressing all of my original comments and those of other reviewers. I have no further suggestions to improve the paper. Thank you.

Author Response

Thank you again for your careful review.

Reviewer 3 Report

The authors addressed all of my comments.  I have no further suggestions. The manuscript was substantially improved and can be now published.

Author Response

(The authors gave the same response as above.)
